# The Functional Profile, Depressive Symptomatology, and Quality of Life of Older People in the Central Alentejo Region: A Cross-Sectional Study

**DOI:** 10.3390/healthcare12222303

**Published:** 2024-11-18

**Authors:** César Fonseca, Bruno Morgado, Elisabete Alves, Ana Ramos, Maria Revés Silva, Lara Pinho, Ana João, Manuel Lopes

**Affiliations:** 1Comprehensive Health Research Centre, University of Évora, 7000-801 Évora, Portugal; cfonseca@uevora.pt (C.F.); elisabete.alves@uevora.pt (E.A.); maria.joao.silva@uevora.pt (M.R.S.); lmgp@uevora.pt (L.P.); alsjoao@uevora.pt (A.J.); mjl@uevora.pt (M.L.); 2Nursing Department, University of Évora, 7000-801 Évora, Portugal; 3LA REAL, Associated Laboratory in Translation and Innovation Towards Global Health, University of Évora, 7000-801 Évora, Portugal; 4Escola de Doctorat, Universitat Rovira i Virgili, 43005 Tarragona, Spain; 5Lisbon School of Nursing, 1600-190 Lisbon, Portugal; ramos.anafilipa@gmail.com; 6Nursing Research, Innovation and Development Centre of Lisbon, 1600-190 Lisbon, Portugal

**Keywords:** older adults, aging, quality of life, depression, functioning

## Abstract

**Background**: Europe’s aging population presents challenges such as a shrinking labor force, pressure on health services, and increased demand for long-term care. This study assesses the functional profile, depressive symptoms, and quality of life of older adults in the Central Alentejo region of Portugal. **Methods**: A cross-sectional, descriptive study was conducted with a convenience sample of 868 older adults in Portuguese long-term care facilities across the Évora district. A structured questionnaire collected sociodemographic data, elderly nursing core set patient information, a health questionnaire with nine responses, and WHO Quality of Life Assessment (short version) scores. **Results**: Nearly half of the participants needed assistance with care. Women (OR = 1.46) and those with cognitive impairment (OR = 10.83) had higher impaired functionality, while education (OR = 0.52) and being overweight (OR = 0.52) were inversely related to functional dependence. Quality of life scores ranged from 56.4 (physical) to 66.6 (environmental). Moderate depressive symptoms were found in 17.1% of participants, with 9% having moderately severe to severe symptoms. Higher dependence doubled the likelihood of depressive symptoms (OR = 2.18). **Discussion and Conclusions**: High rates of depression and functional dependence correlate with a low perception of quality of life, highlighting the need for research to promote and protect the health of older adults.

## 1. Introduction

In recent decades, there has been a remarkable demographic change worldwide, characterized by a substantial increase in the older adult population [1,2]. This phenomenon has become more pronounced in Europe [3,4] and particularly in Portugal [5]. According to the 2021 Portuguese Census, the population aging index was 182, i.e., there are 182 elderly people for every 100 young people [6]. This index significantly increased from 128 in 2011 [6]. The proportion of people of working age (population aged 15 to 64) also fell from 65.8% to 63.1% between 2011 and 2021, while the percentage of elderly people (population aged 65 or over) increased from 19.2% to 24% in the same period, characterizing Portugal as a super-aged society [6]. As the number of citizens over the age of 65 continues to rise, so does the demand for institutions such as residential structures for older adults, commonly known as nursing homes. This growing need for institutionalized care has led to a critical analysis of how technology can be harnessed to improve the quality of life of the people who live there [4].

The world’s population is aging at an unprecedented rate and this demographic transition is fundamentally remodeling societies all over the world [3]. Factors such as increased life expectancy, declining birth rates, and the arrival to “old age” of the post-World War II baby boomer generation have contributed to this phenomenon over the last few decades [3]. According to the World Health Organization (WHO), the global population aged 60 and over is expected to double by 2050, reaching almost 2.1 billion people. This aging trend presents challenges and opportunities for governments, health systems, and communities around the world [4].

Portugal is confronted with distinct challenges associated with its aging population, namely a shrinking labor force, pressure on health services, increased demand for long-term care institutions, and the need for social and economic policies that respond to the needs of older people. This country has one of the highest aging rates in Europe, with a significant proportion of its population aged 65 and older [7]. The life expectancy in Portugal has steadily increased and currently stands at approximately 81 years [7], increasing the pressure on health systems, which results in rising healthcare costs and resource allocation problems [8]. The Central Alentejo region, where this study is focused, is characterized by a higher proportion of older residents compared to other regions, largely due to rural depopulation and younger generations migrating to urban areas [7]. These demographic characteristics make the Central Alentejo region particularly relevant for studying the needs and quality of life of older adults.

As people age, they are often faced with a series of physical and cognitive health problems that can restrict their autonomy [4]. Functional profile refers to the physical, cognitive, and social capabilities that influence an individual’s ability to perform daily activities independently. The functional profile encompasses various dimensions of a person’s health and well-being, including mobility, cognitive functioning, social engagement, and the capacity for self-care. It helps in assessing the extent to which older adults can manage their daily lives autonomously and indicates areas where they may need additional support or intervention [9,10].

This reality leads many individuals and their families to look for long-term institutionalized care options. Residential facilities for older adults provide essential support for those who need assistance with activities of daily living, medical care, and social interaction. However, the growing demand for these institutions presents significant logistical and social problems [9,11,12].

Quality of life (QoL) is defined by the WHO (2019) as an individual’s perception of their position in life within the context of the culture and value systems in which they live, and perceptions of their goals, expectations, standards, and concerns. Thus, this is a multidimensional concept that encompasses physical health, mental well-being, social involvement, and general satisfaction with the environment in which one lives. In this context, providing a high quality of life for residents of long-term care institutions is not only a moral imperative but also a measure of a society’s progress in caring for its older population [13,14,15].

Despite the best efforts of carers, long-term care facilities often face challenges in maintaining and improving residents’ quality of life. The most common problems include loneliness, limited access to personalized care, and a lack of stimulating day-to-day activities [11]. These challenges highlight the need for innovative solutions to improve the well-being of institutionalized individuals. However, to achieve this, it is important to know and map the older institutionalized population in Portugal, to personalize the policies and interventions needed to promote well-being and self-care [16]. A study that was also conducted in the Alentejo region sought to perform a diagnosis of the functionality of institutionalized elderly people in Portalegre city. However, they only focus on the functional part and do not specifically assess the other dimensions of the functionality and quality of life of institutionalized elderly people [7].

This study aims to assess the functional profile, depressive symptoms, and quality of life of older people in the Central Alentejo region, providing a basis for targeted interventions. To guide this research, we pose the following questions: (1) What is the distribution of functionality, quality of life, and depressive symptomatology among older people in the Central Alentejo region? (2) How are the participants’ sociodemographic, clinical, and functional characteristics associated with impaired functionality, perceived quality of life, and depressive symptomatology?

By addressing these gaps, the study seeks to provide a deeper understanding of the aging population in Portugal and inform the development of more effective and personalized care strategies.

## 2. Materials and Methods

A cross-sectional, descriptive study was carried out. The following instruments were applied to a non-probabilistic, convenience sample of 868 older adults, distributed across Portuguese long-term care settings, which include residential structures for older adults, day centers, domiciliary support, and integrated long-term care teams. To this end, institutions were selected from each municipality in the Évora district.

The inclusion criteria were people living in the Alentejo Central region (NUTS III), with health services in residential structures for older adults, day centers, home support services, or integrated long-term care teams; aged 65 or over; and with one of the following medical diagnoses, duly recorded in the medical file: diabetes mellitus, hypertension, COPD or obesity. All people with atrial fibrillation were excluded due to the incompatibility of reading the heart tracing monitoring devices. Twenty-nine institutions in the Évora district agreed to take part in our study, 100% of which have a residential structure for the elderly, 67.07% have a home support service, and 65.52% have a day center.

After obtaining the participants’ informed consent, the data were collected by a team of nurses using the structured interview technique. The nurses who conducted the surveys participated in a detailed training process that included both face-to-face and online sessions focused on the proper use of the assessment instruments. The training covered topics such as the methodology for administering surveys, the interpretation of assessment tools, and protocols for consistent data collection. To ensure uniformity in data collection, they were also provided with a comprehensive guide/manual that outlined step-by-step procedures and clarified any potential doubts, thereby standardizing the data collection process. The average time taken to administer the questionnaire was 20 min. Data collection occurred in an institutional setting between July and October 2023.

The data were processed using the STATA 15.1 statistical program (College Station, TX, USA, 2017). Sample characteristics were presented as counts and proportions. The distribution of the functional profile and the quality of life scores by dimension were presented as means (standard deviation (SD)), while the prevalence of the general functional profile and depressive symptomatology were described as counts and proportions, with the respective 95% confidence intervals (95%CI). To assess the association between sociodemographic and clinical variables with impaired functionality and moderate to severe depressive symptomatology, unconditional binary logistic regression models were fitted to compute crude odds ratios (OR) and 95%CI. Linear regression models were computed to assess the crude association and 95%CI between participants’ sociodemographic, clinical, and functional characteristics and perceived QoL. Statistical significance was set at a value of *p* < 0.05.

### 2.1. Clinical Assessment Instruments

Data were collected on the participant’s age, sex, marital status, educational qualifications, and medical diagnoses. Assessment instruments validated for the Portuguese population and adapted for older people were then used, namely the Elderly Nursing Core Set [17], the Patient Health Questionnaire 9 (PHQ-9) [18], and the World Health Organization’s Quality of Life Assessment Instrument, short version (WHOQOL-Bref) [19], which are described below.

#### 2.1.1. Elderly Nursing Core Set (ENCS)

The ENCS [17] is an instrument that was built on the International Classification of Functioning (ICF) and aims to assess the functionality of older adults. This instrument is made up of five different sections, but only the following sections, related to assessing functionality, were used: II—Body Functions; III—Body Structures; IV—Participation Activities; and V—Environmental Factors. It comprises 31 items which are answered on a 5-point Likert scale, with a positive orientation (No problem: 0–4%; Mild problem: 5–24%; Moderate problem: 25–49%; Severe problem: 50–95%; Complete problem: 96–100%). In terms of its psychometric properties, there is a total explained variance of 82.25%—Kaiser–Meyer–Olkin (KMO) = 0.947, which translates into a high correlation between the various items of the scale. It also shows an overall Cronbach’s alpha of 0.963, revealing excellent reliability for the items presented [17]. Throughout the instrument, four components are assessed, which are subdivided into different codes: Self-care (washing, dressing, caring for parts of the body, moving around using some kind of equipment, walking, carrying out daily routines, maintaining body position, changing basic body position, care related to excretory processes, use of hand and arm, drinking and eating); Learning and Mental Functions (emotional functions, orientation functions, attention functions, memory functions, consciousness functions, and higher-level cognitive functions); Communication (speaking, conversation, communicating and receiving oral messages, and family relationships); and Relationships with friends and carers (personal carers and personal assistants, health professionals, and friends) [17].

#### 2.1.2. Mini-Mental State Examination

Developed by Folstein et al. in 1975 [20] specifically for older adults, this test was translated and adapted to the Portuguese population in 1994 [21]. It determines temporal and spatial orientation, short-term memory (immediate or attention) recall, calculation, coordination of movements, language, and visuospatial skills [21]. It includes eleven items, divided into two sections. The first is for verbal answers to orientation, memory, and attention questions; the second is for reading and writing on naming skills, following verbal and written commands, writing a sentence, and copying a drawing (polygons). All the questions are carried out in the established order and a score is associated with them, totaling the points awarded for each task. In terms of content validity, the test assesses 8 out of 11 main aspects of cognitive state, identifying factors related to orientation, memory, and attention [21]. The score ranges from 0 to 30, and the diagnostic threshold for cognitive deterioration varies depending on the level of schooling: score ≤ 15 for participants without any degree of education, score ≤ 22 for participants with less than 12 years of education, and score ≤ 27 for participants with 12 or more years of education [21]. The internal consistency of this scale is adequate, with Cronbach’s alpha = 0.81. Convergent validity is supported by high correlations and diagnostic accuracy has been adjusted, with a sensitivity of 85.8%, for the Portuguese population [21].

#### 2.1.3. Nine-Item Patient Health Questionnaire (PHQ-9)

This is a self-report instrument that comes from a more comprehensive instrument designed to diagnose psychiatric disorders, The Primary Care Evaluation of Mental Disorders (PRIME-MD) [18]. It has nine items and uses a 4-point response scale ranging from 0 (not at all) to 3 (almost every day) referring to the presence of symptoms in the last two weeks, followed by a functional impact question. This instrument can be used to systematically identify symptoms of depression, and the scores range from 0 to 27. The meanings of the scores obtained are as follows: 0–4 points—no depression; 5–9 points—mild depressive disorder; 10–14 points—moderate depressive disorder; 15–19 points—moderately severe depressive disorder; and 20–27 points—severe depressive disorder. The higher the score, the more severe the depression is [18]. The scale for the Portuguese population has adequate internal consistency, with a Cronbach’s alpha of 0.86, good reliability (ICC = 0.87), and good construct validity, ‘as the overall scores and severity levels were strongly associated with the functional and symptom subscales’ [18].

#### 2.1.4. World Health Organization Quality of Life Instruments-Bref (WHOQOL-Bref)

The World Health Organization Quality of Life Assessment Tool, an abbreviated version the WHOQOL-Bref tool, is designed to assess the quality of life of adult individuals. It is a generic, multidimensional, and multicultural measure for the subjective assessment of the quality of life and can be used across a broad spectrum of psychological and physical disorders, as well as with healthy individuals [19]. The structure of the instrument includes four domains of quality of life: Physical, Psychological, Social Relationships, and Environment. Each of these domains is made up of facets of quality of life that summarize the domain of quality of life in which they fall. This instrument allows for Lickert-type answers (5 points) and can be quoted manually or using statistical software. This measure also makes it possible to calculate an overall indicator, namely the general facet of quality of life. The result should be transformed into a scale from 0 to 100, with a higher final score corresponding to a better perception of quality of life [19]. The Cronbach’s alpha value for the 26 questions in the instrument is 0.91, for the first domain 0.84, for the second 0.78, for the third 0.70, and for the fourth 0.71, which shows satisfactory values both for the questionnaire as a whole and by domain, with the lowest value being for the social relations domain [19].

### 2.2. Ethical Considerations

The principles of dignity, justice, fairness, solidarity, participation, and professional ethics were considered when constructing this project, to eliminate the risk of legal non-compliance or moral offence. The project was timely presented to potential participants (older adults and carers), with a detailed explanation of the objectives. Informed consent was obtained through a process that ensured participants fully understood the voluntary nature of the study, including the right to withdraw at any time without consequences. Additional measures were taken to protect vulnerable participants, such as providing extra time for decision-making, ensuring the presence of a trusted caregiver during the consent process, and regularly checking in with participants to ensure their continued comfort and understanding. The confidentiality and anonymity of the collected data were strictly upheld. Data collected by professionals were coded to protect privacy, and the researchers were responsible for securely storing it on a secure computer platform, further safeguarding participants’ information. The project was submitted to the Ethics Committee of the University of Évora for approval, which was favorable with reference number 22176.

## 3. Results

A total of 868 people from the Alentejo Central region were assessed, with health responses within the framework of services that guarantee social and/or health responses. 

The characteristics of the participants are described in Table 1. Approximately 70% of the carers were female and had an average age of 86.1, ranging from 65 to 104 years old. Many carers were widowed (66.7%) and less than 20% were married or in a civil partnership. More than 50% had attended school but not higher education, while almost 40% had no schooling at all.

High body mass index values were described in this sample, with 37% of participants being overweight and 24.5% being obese. These values are corroborated by the assessment of abdominal obesity, with more than three-quarters of the older adults having waist circumferences above the recommended values. Regarding heart rate, 13.7% of the sample had bradycardia and 1.5% had tachycardia. Tachypnoea was observed in only 3% of the participants, with the remaining having a normal respiratory rate. Almost all the participants reported a diagnosis of two or more chronic illnesses. Cognitive assessment of the older adults was carried out using the Mini Mental State Examination (MMSE), and it was concluded that almost 60% had cognitive deterioration at the time (Table 1).

Figure 1 shows the distribution of the results of the sample’s functional profile, revealing that the highest levels of dependence are found in the Self-care dimension (mean (SD) = 2.6 (1.1)).

The mean value (SD) of the general functional profile (2.3 (1.0)) indicates that around half of the participants need help with care. In fact, 38.4% (95%CI 35.2–41.7) have moderate levels of dependency, while almost a third (31.5%; 95%CI 28.4–34.7) have high levels of dependency requiring daily care (severe problem) or total replacement (complete problem) (Figure 2).

Table 2 shows the participants’ sociodemographic, clinical, and cognitive characteristics associated with impaired functioning, considering the highest level of dependency (moderate, severe, and complete problem). Women (OR = 1.46; 95%CI 1.07–1.99) had a higher prevalence of impaired functioning compared to men. Schooling (OR = 0.52; 95%CI 0.38–0.71), as well as being overweight (OR = 0.52; 95%CI 0.37–0.73), were inversely associated with functional dependence in older adults. Participants with cognitive impairment were 10 times more likely to be dependent on care (OR = 10.83; 95%CI 7.65–15.33) than those without cognitive impairment.

The mean (SD) quality of life ranged from 56.4 (16.8) for the physical dimension to 66.6 (13.8) for the environmental dimension among participants without cognitive impairment. The total mean value (SD) of quality of life was 53.3 (19.5) (Figure 3).

Table 3 describes the association between the participants’ sociodemographic, clinical, cognitive, and functional variables and their perceived quality of life. Women and participants with higher levels of dependency tended to perceive their quality of life more negatively. Conversely, overweight participants more often showed a more positive perception of their quality of life than those with normal weight, although this association was only significant for the psychological (β = 4.66 95%CI 0.30 to 9.01) and environmental (β = 4.14 95%CI 0.23 to 8.06) domains.

The prevalence of depression in older people was assessed by administering the Patient Health Questionnaire-9 (PHQ-9) to all those without cognitive impairment. The mean score (SD) of the questionnaire was 6.5 (5.0), ranging from a minimum of 0 to a maximum of 25 points. In the population assessed, around 75% had minimal or mild signs of depression (74.6%; 95%CI 69.4–79.3), 17.1% (95%CI 13.1–21.8) had moderate symptoms, 7.6% (95%CI 4.9–11.1) had moderately severe symptoms and less than 1% (0.6%; 95%CI 0.1–2.3) had severe depressive symptoms (Figure 4).

There were no statistically significant differences between participants reporting moderate, moderately severe, or severe symptoms about gender, age, marital status, education, and abdominal obesity (Table 4). Overweight people had significantly less depressive symptomatology (OR = 0.47; 95%CI 0.26–0.88) compared to normal weight people. The same trend was observed for obesity, despite the lack of statistical evidence. Participants with a higher level of dependence were twice as likely to describe depressive symptoms (OR = 2.18; 95%CI 1.27–3.81), while quality of life was inversely associated with depression (OR = 0.95; 95%CI 0.94–0.97).

## 4. Discussion

This study offers a description of the functional profile, depressive symptomatology, and quality of life of older adults in Portugal, useful for designing and implementing innovative strategies to improve the health and well-being of institutionalized and non-institutionalized older individuals. A high prevalence of functional dependence and depression among older people, as well as a poor perception of their quality of life, was described. Age and gender inequalities were observed, with older participants and women presenting lower functional levels, higher levels of depression, and a lower perception of quality of life.

### 4.1. Functional Profile of Older Adults

The results reveal that almost all the participants have some problem with their functional profile, which is worrying given the impact of such a result on their independence in carrying out life activities [15,22,23]. Also, the high level of dependence on self-care and learning capacity, and mental functions calls attention to the need to prevent physical and cognitive health problems that frequently restrict their autonomy in the short-, medium-, and long term [23].

Although aging is frequently identified as one of the main causes of loss of muscle mass and physical disability [24], which could justify the worst functional profile among older participants, we found no significant differences according to age in our sample. Women had a higher prevalence of impaired functionality compared to men. These results are in line with several international studies [25,26] arguing that women have a higher prevalence of musculoskeletal pathologies, namely osteoarthritis and osteoporosis, psychiatric pathologies (depression and anxiety) and are also more likely to have reduced visual acuity and incontinence [25,26,27]. Recent studies have also shown an association between low levels of schooling or illiteracy and greater functional incapacity, conclusions that reinforce and confirm our results [28,29]. One possible explanation may be the Portuguese context in the first half of the last century. There was widespread socio-economic hardship, which meant that many children at the time (today’s older adults) had to enter the world of work at a young age, thus taking away the opportunity to attend school. This low socio-economic status, which in many cases was perpetuated throughout life, combined with low literacy, influences lifestyles throughout the life cycle [4].

According to our results, almost all of the participants with impaired functioning also present multimorbidity, i.e., more than one identified medical diagnosis. The association between low levels of functioning in the presence of multimorbidity is supported by the literature, which states that older people over 65 have the worst functional results associated with multimorbidity [30,31,32]. Mobility impairment, whether associated with illness or as a multimorbidity condition, has consequences for the various systems of the human body [33,34]. Other risk factors associated with reduced functioning in older adults are consistent with our results and the published literature: multimorbidity is also associated with age, mainly due to reduced walking speed, but it is also influenced by elements such as not having attended school and being institutionalized [8,27].

Our results suggest that overweight participants had better functional independence, a more positive perception of their quality of life, and low levels of depressive symptomatology. However, adverse anthropometric measures, such as excessive weight and higher waist circumference, are usually described as being associated with worse functionality, social isolation, depression, and the inability to carry out life activities [24]. However, recent data claim that, in some cases, a higher body mass index (BMI) may be associated with functional benefits in older adults, a phenomenon known as the ‘obesity paradox’. This paradox indicates that being overweight, contrary to what is expected, can offer a certain level of protection against conditions of frailty and functional loss, possibly due to greater energy reserves or muscle mass that allows for greater resistance to falls, for example [35]. However, other studies indicate that this association may vary with time and advanced age, suggesting that the benefits of being overweight may be specific to certain age groups and clinical conditions, rather than representing a general rule [36].

Another relevant result of our study is the high level of cognitive deterioration of the participants and its strong association with functional impairment. The literature confirms that older adults with cognitive impairment are more likely to be dependent on care, emphasizing cognition as a fundamental element for functionality. Cognitive stimulation and/or physical activity programs centered on the person can play a fundamental role in preventing and/or delaying the cognitive decline associated with aging, while contributing to promoting the autonomy and functional independence of older adults [33,37,38].

A previous Portuguese study that set out to diagnose the functionality of institutionalized elderly people in the city of Portalegre, a city in the Alentejo region, presented results very similar to ours, corroborating that the sample presents functional deficits, including considered cognitive deficiencies, requiring intervention [7].

### 4.2. Quality of Life of Older Adults

The perception of quality of life was lower than the scores observed in the Portuguese general population [39] aged above 64 years old, especially for the physical and overall domains. This may be justified by the challenges faced by long-term care facilities in maintaining residents’ quality of life [11,40]. Many studies describe a reduction in the perception of quality of life after the institutionalization of the older adult, pointing out the breakdown of daily routines and tastes and, above all, the reduction in the satisfaction of social support, due to the separation from friends and family, as the main reasons for such decrease [41,42,43]. Thus, our results highlight the need to improve the quality of life of older adults after institutionalization in long-term settings.

In comparison to the Portuguese general population, older adults residing in the Alentejo region show a particularly vulnerable perception of quality of life (QoL), which aligns with broader national trends. Studies focusing on this region highlight challenges such as isolation, limited access to health services, and economic difficulties, particularly in rural areas [44].

Internationally, the negative perception of QoL among older adults, especially compared to the general population, is a recognized phenomenon [44]. Studies show that aging populations face higher rates of chronic illness, mobility limitations, and reduced social support, contributing to a diminished QoL [44]. This is consistent across various countries, including those in the European Union, where a significant proportion of older adults experience a decline in physical and social well-being [44].

The results of this study revealed an especially low perception of QoL regarding the physical domain, which may be explained by the old age of our sample and by the fact that many participants reside in long-term care facilities, where access to regular physical activity and specialized care, such as rehabilitation and physical therapy, is often limited [45]. This restrictive environment may contribute to a progressive decline in mobility and physical capacity, negatively affecting individuals’ perception of their physical health and, consequently, their overall quality of life [23]. Therefore, the lack of adequate physical stimulation in these facilities may partially explain the low scores observed, highlighting the importance of integrating rehabilitation practices and physical activities into these settings to promote better quality of life in the physical domain [45].

Women and participants with higher levels of dependency tended to perceive their quality of life more negatively. The current literature supports this conclusion stating that since women have a longer average life expectancy than men, they experience higher rates of multimorbidity and, as previously discussed, lower levels of functionality and self-care, leading to a need for institutionalization in nursing homes, which can result in a more negative perception of their quality of life [8,46,47,48].

Another important result of our study is the tendency observed for an inverse association between quality of life and educational level. Literacy can be understood as a powerful self-care tool, resulting in the improvement in older adults’ capacity for resilience when facing life adversities, which may protect them from the negative meanings of adversity. Thus, it can be partly assumed that increased literacy in older adults can become a protective factor, as previously suggested by the international literature [49,50,51].

### 4.3. Depressive Symptomatology of Older Adults

Depression is the most recurrent psychopathological condition among older people and, due to its high incidence and considerable negative effects on quality of life, it is classified as one of the most relevant geriatric syndromes [16,52]. In our study, depressive symptoms were described in nearly 60% of the older adults assessed. Research indicates that depressive symptoms are common among the elderly in Portugal, with rates varying between 30% and 60%, depending on the specific population studied. In rural regions like Alentejo, isolation and limited access to health and social services can exacerbate these rates [44]. Internationally, the prevalence of depressive symptoms in older adults also reflects similar trends. For instance, studies across Europe and the OECD report rates ranging from 20% to 50%, often influenced by factors such as chronic illness, institutionalization, and lack of social support [44]. Thus, the nearly 60% prevalence in our study is within the expected range based on both national and international data.

Our data intersect with some of the most commonly identified risk factors: female, widowed, polymedication, low literacy, low socio-economic status, unemployment, compromised physical health (multimorbidity), loneliness, lack of social support, stressful life events, nutritional deficits, cognitive impairment, and neurodegenerative diseases [16,53]. Despite the lack of statistical significance, a tendency for women to present higher depressive symptomatology can be observed. Gender inequalities may be explained by the role of Portuguese women in the first half of the last century. Women were traditionally responsible for taking care of the home, children, husband, and family. However, the departure of children from the home, the death of family members, and finally the death of the husband may result in a total loss of purpose and social support, culminating in the emergence of depressive symptoms in older Portuguese women [31,32]. The association between depression and functionality is also supported by the previous literature, with a recent international study concluding that 27% of older people have depression, the most common condition among older adults, with a relevant impact on their autonomy and quality of life [54]. In this context, a person-centered care process is therefore essential if real health gains are to be achieved for Portuguese older people with depression [1,23,55,56].

### 4.4. Strengths and Limitations of the Study

A key methodological advantage of the present study is the consecutive and systematic data collection over 4 months, by trained nurses, ensuring data quality control. However, some limitations should be noticed. The cross-sectional nature of our study does not allow the establishment of causality between any of the characteristics assessed and functionality, quality of life, and depression. Furthermore, the focus on the Alentejo region hinders the representativeness of the sample, thus characterizing only the population of the interior of Portugal. Also, a convenience sampling was used, which may reduce the generalizability of the findings to a broader population. However, the relatively large sample size, which was calculated to ensure a power of 80% to demonstrate associations with a magnitude of at least 1.5 (odds ratio) at a 5% significance level, as well as the inclusion of 29 institutions in the Alentejo region ensures the robustness of our sample. Still, future research should use probabilistic or random sampling techniques, with larger samples and covering a broader geographic area to ensure a more representative sample. This will enhance the validity and applicability of the results across different cultures and contexts.

In the present study, we have included participants of 65 or more years in age, distributed across different Portuguese long-term care settings. Such an approach allowed us to incorporate both dependent individuals (living in care homes) and more independent older adults (participating in day centers), ensuring a diverse representation rather than focusing on a single profile of older individuals. Although we may acknowledge the possibility of selection bias, it is important to emphasize that, in each institution, all participants who met the selection criteria were invited to participate, and we had very few refusals. Also, although there were missing data for some variables, the maximum percentage of missing data was low (less than 5% for all variables, except for multimorbidity, which was nearly 10%). Thus, it is not expected to have a significant impact on the main results presented, with the recent literature supporting that approximately 5% of missing data is acceptable and can be ignored [57,58].

The lack of a significant correlation between age and functionality, previously described in the literature, can be explained by the large proportion of older adults aged above 80 years included in our sample, which results in small variations in the age of the participants. Thus, the lack of discrimination between different age groups may justify our results. This phenomenon is described in various studies analyzing functionality in elderly populations, where it is observed that the variability of factors such as mobility, cognition and physical capacity tends to stabilize or be influenced by other factors than chronological age [59,60]. Despite the absence of statistical significance in the association between depression and the female gender, justified by the gender disproportion in the sample, the results show a tendency for women to present more depressive symptoms than men. Several previous studies have shown that the female gender is associated with a higher prevalence of depression, a phenomenon that has been widely documented, especially in large-scale epidemiological studies [61]. Also, a systematic review confirmed that women have twice the prevalence of depression compared to men, mainly due to hormonal, biological, and psychosocial factors [62]. However, the low proportion of men in our sample probably limits the statistical power of the study to detect a statistically significant association between sex and depressive symptomatology.

## 5. Conclusions

The main results of this paper emphasize the need to design and implement future interventions to promote quality of life, independence, and empowerment of older citizens at a physical, cognitive, emotional, and social level. Also, extending the assessment of functional profiles, mental health, and perception of quality of life to the national territory to obtain a representative sample of Portugal would contribute to the establishment of national guidelines for improving health and general well-being. It is therefore essential to invest in innovative healthcare models, supported by health information technologies, that promote healthy lifestyles, foster the creation and maintenance of social support networks, facilitate access to specialized medical care, and contribute to the early detection of health warning signs and symptoms. A change in approach and intervention is therefore required, according to three main pillars: (1) giving the citizen centrality, taking the focus of decision-making away from professionals and refocusing the entire care process on the person as their manager; (2) investing in people’s literacy by empowering them to access, understand, and use health information for their management process; and (3) improving the integration and continuity of care, so that health services are easy to navigate and each person can easily find the care they need when they need it [5,8,17,63,64].

In conclusion, this study reinforces the need to design and implement multidisciplinary research projects aimed at promoting, preventing, and protecting older adults’ health. These initiatives will benefit older adults, and, simultaneously, contribute to building a healthier and more resilient society.

## Figures and Tables

**Figure 1 healthcare-12-02303-f001:**
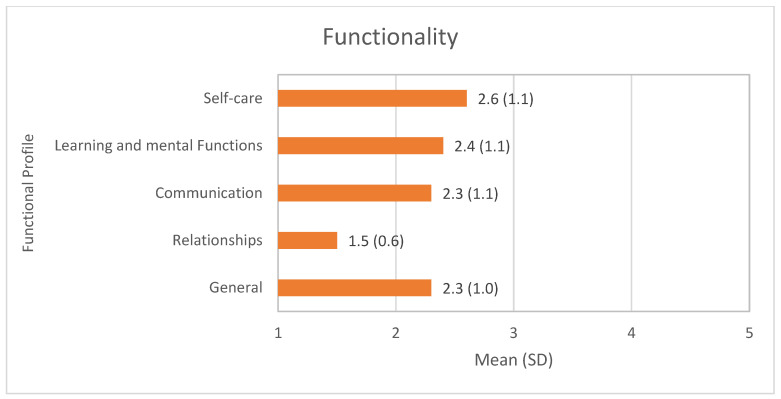
Functioning profile by domains of the Elderly Nursing Core Set (ENCS).

**Figure 2 healthcare-12-02303-f002:**
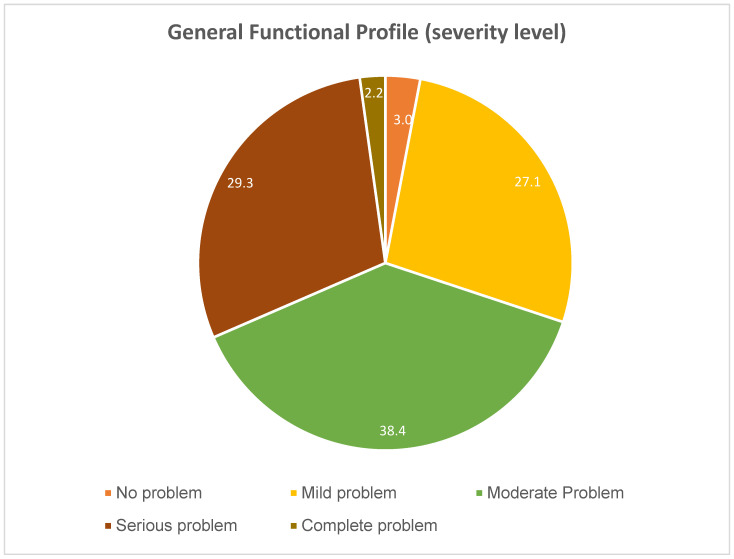
General functional profile assessed using the Elderly Nursing Core Set (ENCS).

**Figure 3 healthcare-12-02303-f003:**
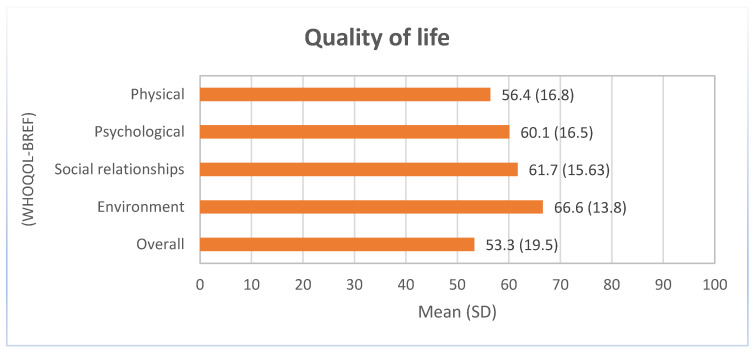
Distribution of the quality of life score by dimension of the World Health Organization’s quality of life assessment tool (WHOQOL-Bref).

**Figure 4 healthcare-12-02303-f004:**
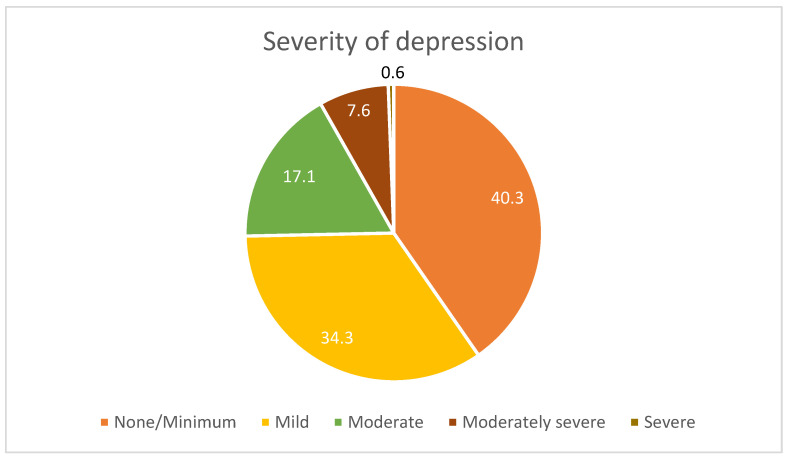
Severity of depressive symptoms assessed by the Patient Health Questionnaire-9 (PHQ-9).

**Table 1 healthcare-12-02303-t001:** Sociodemographic characteristics of the sample.

Variable	Description	*n*	%
Sex	Male	257	29.7
	Female	607	70.3
Age (years)	Mean (min–max)	86.1 (65–104)	
Marital status	Single	100	11.6
	Married/cohabiting	169	19.6
	Widowed	576	66.7
	Divorced/separated	19	2.2
Educational level	Did not go to school and cannot read or write	326	37.7
	Did not go to school, but can read and write	45	5.2
	Attended school but not higher education	483	55.9
	Attended higher education	10	1.2
Body mass index (kg/m^2^)	<25.0	326	38.5
	25.0–29.9	313	37.0
	≥30.0	207	24.5
Abdominal circumference *	No	200	23.2
	Yes	662	76.8
Heart rate	Bradycardia	119	13.7
	Normal frequency	736	84.8
	Tachycardia	13	1.5
Respiratory rate	Bradypnea	0	0.0
	Normal frequency	842	97.0
	Tachypnea	26	3.0
Multimorbidity **	No	2	0.3
	Yes	747	99.7
Cognitive impairment (MMSE)	No	328	40.4
	Yes	483	59.6

The total does not add up to 868 in all variables since not all the study subjects answered all the questions. * Waist circumference > 88 cm in women and >102 cm in men; ** Diagnosis of 2 or more chronic diseases.

**Table 2 healthcare-12-02303-t002:** Association between participants’ sociodemographic, clinical, and cognitive variables and impaired functioning.

Variable	Description	Moderate, Severe, or Complete Problem			
		*n*	(%)	Crude OR	95%CI
Sex	Male	164	64.3	1	
	Female	438	72.4	**1.46**	**1.07–1.99**
Age (years)	65–79	82	73.9	1	
	≥80	502	69.5	0.81	0.51–1.27
Marital status	Married/cohabiting	111	65.7	1	
	Single, widowed, or divorced/separated)	491	71.01	1.28	0.90–1.83
Educational level	Did not attend school	286	77.5	1	
	Attended school	315	64.2	**0.52**	**0.38–0.71**
Body mass index (kg/m^2^)	<25.0	250	76.7	1	
	25.0–29.9	195	63.1	**0.52**	**0.37–0.73**
	≥30.0	145	70.1	0.71	0.48–1.05
Abdominal circumference *	No	150	75.4	1	
	Yes	450	68.3	0.70	0.49–1.01
Multimorbidity, *n* (%) **	No	2	100.0	- -	
	Yes	516	69.4		
Cognitive impairment (MMSE)	No	144	42	1	
	Yes	457	88.7	**10.83**	

OR, Odds Ratio; 95%CI, 95% Confidence Interval; bold indicates statistically significant associations (*p* < 0.05); * waist circumference > 88 cm in women and >102 cm in men; ** Diagnosis of 2 or more chronic diseases.

**Table 3 healthcare-12-02303-t003:** Association between participants’ sociodemographic, clinical, and functional variables and perceived quality of life.

		Quality of Life									
Variable	Description	Overall		Physical		Psychological		Social Relationships		Environment	
		Crude β	95%CI	Crude β	95%CI	Crude β	95%CI	Crude β	95%CI	Crude β	95%CI
Sex	Female vs. Male	−0.08	−4.57 to 4.42	**−4.33**	**−8.13** to **−0.53**	**−5.55**	**−9.29** to **−1.82**	−2.22	−5.74 to 1.30	−1.95	−5.34 to 1.44
Age (years)		0.24	−0.13 to 0.61	−0.22	−0.53 to 0.109	0.09	−0.40 to 0.23	0.02	−0.27 to 0.31	−0.05	−0.33 to 0.23
Marital Status	Single, widowed, or divorced/separated vs. married/in a civil partnership	−0.67	−5.92 to 4.57	1.79	−2.67 to6.25	−3.93	−8.32 to 0.45	−2.30	−6.40 to 1.81	0.80	−3.17 to 4.76
Educational level	Educational level	0.14	−4.27 to 4.55	−0.53	−4.28 to 3.22	0.72	−3.00 to 4.44	−0.23	−3.69 to 3.23	−0.61	−3.95 to 2.72
Body mass index (kg/m^2^)	25.0–29.9 vs. <25.0	3.83	−1.40 to 9.06	4.42	−0.02 to 8.85	**4.66**	**0.30** to **9.01**	3.95	−0.11 to 8.01	**4.14**	**0.23** to **8.06**
	≥30.0 vs. <25.0	−0.49	−6.34 to 5.36	0.48	−4.48 to 5.44	2.02	−2.85 to 6.89	1.49	−3.05 to 6.03	0.92	−3.6 to 5.31
Abdominal circumference *	Yes vs. No	−0.81	−6.29 to4.67	−0.88	−5.951 to 3.74	−0.49	−5.08 to 4.10	1.60	−2.70 to 5.91	0.81	−3.27 to 4.89
Multimorbidity **	Yes vs. No	-	-	-	-	-	-	-	-	-	-
Functionality (ENCS)	Moderate, severe, or complete problem vs. No or slight problem	**−8.01**	**−12.36** to **−3.66**	**−12.51**	**−16.05** to **−8.96**	**−8.01**	**−11.65** to **−4.37**	**−8.29**	**−11.65** to **−4.93**	**−7.34**	**−10.61** to **−4.07**

95% CI, 95% Confidence Interval; bold indicates statistically significant associations (*p* < 0.05). * Waist circumference > 88 cm in women and >102 cm in men; ** diagnosis of 2 or more chronic diseases.

**Table 4 healthcare-12-02303-t004:** Association between sociodemographic variables, clinical functioning, and perceived quality of life of participants and depressive symptoms.

Variable	Description	Moderate, Moderately Severe, or Severe Symptomatology			
		*n*	%	Crude OR	95%CI
Sex	Male	25	21.0	1	
	Female	54	28.0	1.46	0.85–2.51
Age (years)	65–79	5	16.7	1	
	≥80	73	26.8	1.83	0.68–4.97
Marital Status	Married/cohabiting	19	27.1	1	
	Single, widowed, or divorced/separated	60	24.9	0.89	0.49–1.63
Educational level	Did not attend school	35	25.6	1	
	Attended school	44	25.1	0.98	0.59–1.64
Body mass index (kg/m^2^)	<25.0	32	32.7	1	
	25.0–29.9	23	18.7	**0.47**	**0.26–0.88**
	≥30.0	20	25.3	0.70	0.36–1.35
Abdominal circumference *	No	17	27.59	1	
	Yes	61	24.4	0.84	0.45–1.57
Multimorbidity **	No	0	0.0	-	-
	Yes	70	25.5	-	-
Functionality (ENCS)	No problem or slight	36	19.0	1	
	Moderate, severe, or complete problem	43	34.8	**2.27**	**1.35–3.81**
General quality of life, mean (SD)		50.6	17.4	**0.95**	**0.94–0.97**

OR, Odds Ratio; 95%CI, 95% Confidence Interval; bold indicates statistically significant associations (*p* < 0.05). * Waist circumference > 88 cm in women and >102 cm in men; ** diagnosis of 2 or more chronic diseases.

## Data Availability

Data is contained within the article.

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
