# Peer review of "The Functional Profile, Depressive Symptomatology, and Quality of Life of Older People in the Central Alentejo Region: A Cross-Sectional Study"

_healthcare, 2024, doi:10.3390/healthcare12222303_

Round 1
Reviewer 1 Report (Previous Reviewer 1)
Comments and Suggestions for Authors
This paper provides a valuable analysis, reflecting the current situation in Portugal and utilizing a broad sample, making it a significant contribution to the field.
Introduction
The study begins by addressing the increasing elderly population and the associated challenges. However, it would benefit from a more detailed explanation of the aging population in Portugal. While the paper mentions the rising elderly population, it would be helpful to provide specific statistics on the percentage of the population aged 65 and over. For instance, if more than 20% of the total population is over 65, the country could be classified as a super-aged society. Clarifying whether Portugal is already in this category or approaching it would enhance the paper. Additionally, it would be useful to clearly define the research questions at the end of the introduction.
Materials and Methods
The Cronbach's alpha values for the instruments used are well presented, and the tools themselves appear to be widely used in previous research, lending credibility to the methodology. However, it should be noted that the sample was selected using a convenience sampling method, which should be addressed as a limitation in the discussion section.
Results
The tables (Table 1 to Table 4) should be presented following APA style guidelines for improved readability. Currently, they lack clarity. Furthermore, the results indicate that overweight participants showed better functional independence and quality of life. This finding should be discussed in depth in the discussion section. However, it appears that this point was not explored adequately in the current discussion.
Discussion
One of the limitations of this study is the use of a convenience sample, which may affect the generalizability of the results. It should be clearly stated that the study may not fully represent the elderly population of Portugal. Additionally, the discussion should suggest directions for future research, such as conducting studies with larger, randomized samples or studies covering a broader geographic area.
In the discussion section, the reason for the low quality of life scores in the physical domain should be elaborated. While the results show that the physical domain had the lowest scores, the discussion does not provide an in-depth analysis of this finding. Explaining the reasons for the low scores in the physical domain would enrich the interpretation of the results. One potential explanation could be that the participants reside in long-term care facilities, which may limit their physical activity.
Author Response
This paper provides a valuable analysis, reflecting the current situation in Portugal and utilizing a broad sample, making it a significant contribution to the field.
Authors: We thank the reviewer for taking the time to carefully review our article.
- Introduction
The study begins by addressing the increasing elderly population and the associated challenges. However, it would benefit from a more detailed explanation of the aging population in Portugal. While the paper mentions the rising elderly population, it would be helpful to provide specific statistics on the percentage of the population aged 65 and over. For instance, if more than 20% of the total population is over 65, the country could be classified as a super-aged society. Clarifying whether Portugal is already in this category or approaching it would enhance the paper. Additionally, it would be useful to clearly define the research questions at the end of the introduction.
Authors: As suggested by the reviewer we are now providing a more detailed overview of the age structure in Portugal and the context of demographic ageing, highlighting that, in 2021, 24% of the population was aged 65 or over (please see page 1, lines 36 to 41. In addition, at the end of the introduction, we reinforced the research questions more clearly and concisely (please see page 3, lines 100 to 107).
- Materials and Methods
The Cronbach's alpha values for the instruments used are well presented, and the tools themselves appear to be widely used in previous research, lending credibility to the methodology. However, it should be noted that the sample was selected using a convenience sampling method, which should be addressed as a limitation in the discussion section.
Authors: We thank the reviewer for his appreciation. As suggested, a paragraph discussing the convenience sampling method used has been added to the limitations of the study (please see page 17, lines 506 to 522).
- Results
The tables (Table 1 to Table 4) should be presented following APA style guidelines for improved readability. Currently, they lack clarity. Furthermore, the results indicate that overweight participants showed better functional independence and quality of life. This finding should be discussed in depth in the discussion section. However, it appears that this point was not explored adequately in the current discussion.
Authors: We thank the reviewer for calling our attention to these issues. We have standardize our tables according to APA guidelines and have also reviewed multiple recently published papers in Healthcare to ensure that all tables comply with the journal's publication criteria (please see Tables 1-4). Additionally, a more in-depth discussion of the association between being overweight and better functional independence, increased perception of quality of life, and lower levels of depressive symptomatology was also introduced in the article discussion (please see page 16, lines 434 to 445).
- Discussion
One of the limitations of this study is the use of a convenience sample, which may affect the generalizability of the results. It should be clearly stated that the study may not fully represent the elderly population of Portugal. Additionally, the discussion should suggest directions for future research, such as conducting studies with larger, randomized samples or studies covering a broader geographic area.
Authors: As previously stated, a paragraph discussing the convenience sampling method used has been added to the limitations of the study. Also the need to perform future research using probabilistic or random sampling techniques, with larger samples and covering a broader geographic area to ensure a more representative sample was enphazised, contributing to enhance the validity and applicability of the results across different cultures and contexts (please see page 17, lines 510 to 514).
5) In the discussion section, the reason for the low quality of life scores in the physical domain should be elaborated. While the results show that the physical domain had the lowest scores, the discussion does not provide an in-depth analysis of this finding. Explaining the reasons for the low scores in the physical domain would enrich the interpretation of the results. One potential explanation could be that the participants reside in long-term care facilities, which may limit their physical activity.
Authors: We thank the reviewer for calling our attention to this point. We have indeed made this adjustment and expanded the discussion on the reasons for the low scores in the physical quality of life domain. For the revised section, please see page 16, lines 434 to 445.
Reviewer 2 Report (Previous Reviewer 2)
Comments and Suggestions for Authors
Thank you for the opportunity to review the authors' paper.
I would like to make the following corrections
The description of L239 in the text is “average age (SD) of 86.1 (6.7)”, but the description in Table 1 is “Mean (min-máx) 86.1 (65-104)”. It should be unified.
In Table 2, the position of 95% CI for “Attended school” is out of alignment with other items.
Only the 95% CI for YES in “Cognitive impairment (MMSE)” is on two lines.
Please unify them.
The position of the numbers in References is misaligned.
For example, 3, 5, 7, 10, etc.
Please unify them.
That is all.
Author Response
Thank you for the opportunity to review the author’s paper.
Authors: We thank the reviewer for taking the time to carefully review our article.
I would like to make the following corrections
1) The description of L239 in the text is “average age (SD) of 86.1 (6.7)”, but the description in Table 1 is “Mean (min-máx) 86.1 (65-104)”. It should be unified.
Authors: We have standardized the text according to the data described in the table (please see page 5, line 245).
2) In Table 2, the position of 95% CI for “Attended school” is out of alignment with other items.
Only the 95% CI for YES in “Cognitive impairment (MMSE)” is on two lines.
Please unify them.
Authors: We have corrected the data in Table 2 accordingly. Please see page 9, Table 2.
3) The position of the numbers in References is misaligned.
For example, 3, 5, 7, 10, etc.
Please unify them.
Authors: We have corrected and aligned the references. Please see page 20 lines 618 to 637.
That is all.
Authors: Thank you for all the corrections and suggestions.
This manuscript is a resubmission of an earlier submission. The following is a list of the peer review reports and author responses from that submission.
Round 1
Reviewer 1 Report
Comments and Suggestions for Authors
I have thoroughly reviewed your manuscript, which presents a significant study on the functional ability, depression, and quality of life among institutionalized older adults in Portugal. The research is noteworthy for its large sample size and the use of well-established assessment tools, which contribute to the reliability of the findings. The focus on these three critical factors affecting older adults' well-being is both intriguing and potentially impactful for future applications.
Introduction:
The study's focus on a specific region in Portugal is interesting. However, it would be beneficial to provide more context on why this particular area was chosen. For instance, elaborating on the severity of aging issues in this region would strengthen the rationale for your study.
The literature review could be more comprehensive. A broader examination of similar studies would help to highlight the unique contributions of your research and position it within the existing body of knowledge.
I suggest clearly stating your research questions or hypotheses to guide the reader's understanding of your study's objectives.
It would be helpful to provide a clear definition and explanation of the term "functional profile" as it is a key concept in your study.
Materials and Methods:
While the typical definition of older adults includes those 65 and above, your study includes individuals from 50 years old. A justification for this inclusion criterion would be valuable.
Please provide more details on the training process for the nurses who conducted the surveys and the measures taken to ensure consistency in data collection.
The ethical considerations section could be expanded. More information on the consent process and any additional measures taken to protect vulnerable participants would strengthen this section.
Information on the reliability and validity of the measurement tools, particularly for the Portuguese versions, would enhance the methodological rigor of your study.
It would be beneficial to explain how each statistical technique was employed to address specific research hypotheses.
A description of how missing data were handled would be a valuable addition to this section.
Results:
I recommend presenting your results in tables following APA format for improved clarity and readability.
Including confidence intervals would provide a better understanding of the precision of your results.
Discussion:
To highlight the significance of your findings, consider comparing your results with similar studies conducted in other regions of Portugal or in different countries.
Some of your results may have been unexpected. A more in-depth discussion of these findings, supported by comparisons with similar studies, would enrich this section.
Overall, your study provides valuable insights into the well-being of institutionalized older adults. Addressing these suggestions could further enhance the impact and clarity of your research. Thank you for the opportunity to review this important work.
Author Response
1) I have thoroughly reviewed your manuscript, which presents a significant study on the functional ability, depression, and quality of life among institutionalized older adults in Portugal. The research is noteworthy for its large sample size and the use of well-established assessment tools, which contribute to the reliability of the findings. The focus on these three critical factors affecting older adults' well-being is both intriguing and potentially impactful for future applications.
Authors: We thank the reviewer for taking the time to carefully review our article.
2) Introduction
The study's focus on a specific region in Portugal is interesting. However, it would be beneficial to provide more context on why this particular area was chosen. For instance, elaborating on the severity of aging issues in this region would strengthen the rationale for your study.
The literature review could be more comprehensive. A broader examination of similar studies would help to highlight the unique contributions of your research and position it within the existing body of knowledge.
I suggest clearly stating your research questions or hypotheses to guide the reader's understanding of your study's objectives.
It would be helpful to provide a clear definition and explanation of the term "functional profile" as it is a key concept in your study.
Authors: We would like to express our gratitude to the reviewer for their valuable comments that have significantly enhanced the clarity and quality of our introduction section. Specifically, we have expanded our discussion on the key characteristics of Portugal, particularly the Alentejo region, to underscore the importance of focusing on the assessment of functionality, depression, and quality of life among older adults (please see page 2, lines 47-58). Furthermore, we have incorporated a paragraph highlighting the primary findings of previous literature in this area and underscoring the unique contribution of our study (please see page 2, lines 86-90). In response to the reviewer's suggestion, we have included a clear definition and explanation of the term "functional profile" (please see page 2, lines 62-66). Lastly, we have introduced our research questions at the end of the introduction section (please see pages 2-3, lines 95-98).
3) Materials and Methods:
While the typical definition of older adults includes those 65 and above, your study includes individuals from 50 years old. A justification for this inclusion criterion would be valuable.
Authors: We acknowledge the reviewer's point about our inconsistency with the current literature regarding the definition of older adults. Therefore, and given the small number of participants below 65 years old (n=27; 3.1% of the total sample), we have decided to exclude them from the analysis and focus solely on participants aged 65 years old or more. All tables and the main text have been revised to reflect this exclusion (please see Results section and Tables 1 to 4).
4) Please provide more details on the training process for the nurses who conducted the surveys and the measures taken to ensure consistency in data collection.
Authors: We thank the reviewer for this comment and additional data on the training process for the nurses who conducted the surveys and the measures taken to ensure consistency in data collection are now being reported in materials and methods section (please see page 3, lines 118-125): “The nurses who conducted the surveys, participated in a detailed training process that included both face-to-face and online sessions focused on the proper use of the assessment instruments. The training covered topics such as the methodology for administering surveys, the interpretation of assessment tools, and protocols for consistent data collection. To ensure uniformity in data collection, they were also provided with a comprehensive guide/manual that outlined step-by-step procedures and clarified any potential doubts, thereby standardizing the data collection process.”
5) The ethical considerations section could be expanded. More information on the consent process and any additional measures taken to protect vulnerable participants would strengthen this section.
Authors: Taking into account the reviewer suggestion, the ethical considerations section was expanded: “The project was timely presented to potential participants (older adults and carers), with a detailed explanation of the objectives. Informed con-sent was obtained through a process that ensured participants fully understood the voluntary nature of the study, including the right to withdraw at any time without consequences. Additional measures were taken to protect vulnerable participants, such as providing extra time for decision-making, ensuring the presence of a trusted caregiver during the consent process, and regularly checking in with participants to ensure their continued comfort and understanding. The confidentiality and anonymity of the collected data were strictly upheld. Data collected by professionals were coded to protect privacy, and the researchers were responsible for securely storing it on a secure computer platform, further safeguarding participants' information.” (please see page 5, lines 219-229).
6) Information on the reliability and validity of the measurement tools, particularly for the Portuguese versions, would enhance the methodological rigor of your study.
Authors: According to the reviewer suggestion, information on the reliability and validity of the measurement tools, particularly for the Portuguese versions, were added to material and methods section (please see pages 4, lines 153-157; pages 4, lines 182-184; pages 4. Lines 195-198 and page 5, lines 211-215).
7) It would be beneficial to explain how each statistical technique was employed to address specific research hypotheses.
Authors: We thank the reviewer for the comment. We are now clearly stating how each statistical technique was employed to answer the research hypotheses previously defined (please see page 3, lines 127-138).
8) A description of how missing data were handled would be a valuable addition to this section.
Authors: For some variables it was not possible to collect data from all participants and some missing data was present. Missing data analysis was not performed due to its low proportion (less than 5% in all variables, except multimorbidity, which was near 10%); thus, we believe that such a limited number of missings will not have a considerable impact on the main results presented. In fact, literature supports that nearly 5% of missing data is acceptable and can be ignored (Heymans & Twisk, 2022; Schafer, 1999). However, we agree with the reviewer and are aware that missing data is important to be considered and discussed, a paragraph was added to study limitations, discussing the potential implications of missing data (please see page 14, lines 457-461).
9) Results:
I recommend presenting your results in tables following APA format for improved clarity and readability.
Authors: (please see tables 1 to 4).
10) Including confidence intervals would provide a better understanding of the precision of your results.
Authors: Tables 2, 3 and 4 are displaying 95% Confidence Intervals (95%CI) for all the estimates presented. Additionally, we are presenting now 95%CI for the prevalence of the general functional profile and depressive symptomatology (please see page 8, lines 286-291, page 10, lines 306-310 and page 11, lines 322-327, respectively).
11) Discussion:
To highlight the significance of your findings, consider comparing your results with similar studies conducted in other regions of Portugal or in different countries.
Authors: As suggested, the significance of our findings was highlighted, through the comparison of our results with similar studies conducted in other regions of Portugal or in different countries (please see page 15, lines 389-392; lines 403-413; lines 430-438).
12) Some of your results may have been unexpected. A more in-depth discussion of these findings, supported by comparisons with similar studies, would enrich this section.
Authors: A more in-depth analysis of unexpected results and their potential justifications was included in the discussion section of the paper (please see page 17, lines 469-486).
13) Overall, your study provides valuable insights into the well-being of institutionalized older adults. Addressing these suggestions could further enhance the impact and clarity of your research. Thank you for the opportunity to review this important work.
Authors: The authors would like to thank the reviewer for the issues raised in the comments, which were helpful to improve and clarify our work.
Reviewer 2 Report
Comments and Suggestions for Authors
Thank you for the opportunity to serve as a reviewer.
I have carefully read your manuscript.
I have some major concerns.
Please see below for details.
About the title and keywords
The title uses "older people" and the keywords "older adults" and "elderly". I think it would be better to standardize the terms used. The same goes for the text.
Regarding the introduction
Readers would be able to better understand the background of the study if you could briefly present Portugal's population, aging rate, life expectancy, and social security system.
A similar explanation is also needed for the Central Alentejo region.
Regarding materials and methods
The selection criterion for subjects was 50 years or older. Is 50 years old considered old? Table 1 also states that the subjects are between 45 and 104 years old. This includes people younger than 50 years old, which creates a discrepancy. In addition, lumping such an age range together creates a sampling bias.
Please indicate the proportion of subjects in each of "residential structures for older adults", "day centers", "home support services" and "integrated long-term care teams". Are there any differences in the amount of support needed by the subjects? Again, I am concerned about sampling bias.
Please explain the selection criteria for the clinical assessment instruments in this study.
If these points are not made clear, it will be difficult to determine the consistency of the results and discussion presented by the authors.
Regarding the results
Please present charts and graphs in color.
This will help readers to better understand.
That's all.
Author Response
1) Thank you for the opportunity to serve as a reviewer. I have carefully read your manuscript. I have some major concerns. Please see below for details.
Authors: We thank the reviewer for taking the time to carefully review our article. The paper was reviewed and we believe that the changes made addressed well all the reviewer’s concerns and suggestions.
2) About the title and keywords:
The title uses "older people" and the keywords "older adults" and "elderly". I think it would be better to standardize the terms used. The same goes for the text.
Authors: We agree with the suggestion and have, therefore, removed the keyword “elderly” because it is no longer used in scientific literature. However, please note that the word "elderly" is still used when reporting the name of one of the instruments used (Elderly Nursing Core Set) and in the references section.
3) Regarding the introduction:
Readers would be able to better understand the background of the study if you could briefly present Portugal's population, aging rate, life expectancy, and social security system. A similar explanation is also needed for the Central Alentejo region.
Authors: As suggested, we have expanded our discussion on the key characteristics of Portugal, particularly the Alentejo region, to underscore the importance of focusing on the assessment of functionality, depression, and quality of life among older adults (please see page 2, lines 47-58).
4) Regarding materials and methods:
The selection criterion for subjects was 50 years or older. Is 50 years old considered old? Table 1 also states that the subjects are between 45 and 104 years old. This includes people younger than 50 years old, which creates a discrepancy. In addition, lumping such an age range together creates a sampling bias.
Authors: Authors: We acknowledge the reviewer's point about our inconsistency with the current literature regarding the definition of older adults. Therefore, and given the small number of participants below 65 years old (n=27; 3.1% of the total sample), we have decided to exclude them from the analysis and focus solely on participants aged 65 years old or more. All tables and the main text have been revised to reflect this exclusion (please see Results section and Tables 1 to 4).
5) Please indicate the proportion of subjects in each of "residential structures for older adults", "day centers", "home support services" and "integrated long-term care teams". Are there any differences in the amount of support needed by the subjects? Again, I am concerned about sampling bias.
Authors: Data was collected from several Portuguese long-term care settings located in the Évora district, including Residential Structures for older adults, Day Centers, Domiciliary support, and integrated long-term care teams, in a total of 29 institutions. However, their individual results are not shown to ensured that the data remain anonymous and non-identifiable. Please see page 17 Lines 487 to 498
6) Please explain the selection criteria for the clinical assessment instruments in this study.
Authors: All the assessment instruments used (Elderly Nursing Core Set, Patient Health Question-naire 9 (PHQ-9) and the World Health Organization’s Quality of Life Assessment Instrument, short version (WHOQOL-Bref) were previously validated for the Portuguese population and adapted for older people. Additional information on the reliability and validity of the measurement tools, particularly for the Portuguese versions, were added to material and methods section (please see pages 4, lines 151-155; pages 4, lines 180-182; pages 4. Lines 193-196 and page 5, lines 209-213).
7) If these points are not made clear, it will be difficult to determine the consistency of the results and discussion presented by the authors.
Authors: We believe that the changes made addressed all the reviewer concerns and suggestions.
8) Regarding the results:
Please present charts and graphs in color. This will help readers to better understand.
Authors: We thank the reviewer for calling our attention to this point. To improve the readers’ readability all charts and graphs are now being presented in color (please see Figures 1 to 4).
Round 2
Reviewer 2 Report
Comments and Suggestions for Authors
I believe an appropriate correction has been made.
Is the "95% IC" listed in Table 2-4 a misprint for "95% CI"?
Please confirm.
That is all.
Author Response
1) I believe an appropriate correction has been made.
Authors: We thank the reviewer for taking the time to carefully review our article.
2) Is the "95% IC" listed in Table 2-4 a misprint for "95% CI"? Please confirm.
Authors: We have corrected the misprint (Please see tables 2-4).